# Linear attention is (maybe) all you need
# (to understand Transformer optimization)

**Kwangjun Ahn**[⋆]
MIT EECS/LIDS
kjahn@mit.edu

**Xiang Cheng**[⋆]
MIT LIDS
chengx@mit.edu

**Minhak Song**[⋆]
KAIST ISysE/Math
minhaksong@kaist.ac.kr

**Chulhee Yun**
KAIST AI
chulhee.yun@kaist.ac.kr

**Ali Jadbabaie**
MIT CEE/LIDS
jadbabai@mit.edu

**Suvrit Sra**
MIT EECS/LIDS
suvrit@mit.edu

## Abstract

Transformer training is notoriously difficult, requiring a careful design of optimizers and use of various heuristics. We make progress towards understanding the subtleties of training Transformers by carefully studying a simple yet canonical linearized *shallow* Transformer model. Specifically, we train linear Transformers to solve regression tasks, inspired by J. von Oswald et al. (ICML 2023), and K. Ahn et al. (NeurIPS 2023). Most importantly, we observe that our proposed linearized models can reproduce several prominent aspects of Transformer training dynamics. Consequently, the results obtained in this paper suggest that a simple linearized Transformer model could actually be a valuable, realistic abstraction for understanding Transformer optimization.

## 1 Introduction

Transformer architectures (Vaswani et al., 2017) (henceforth, referred to as *Transformers*) have shown impressive performance in various applications (Devlin et al., 2019; Bubeck et al., 2023). However, training Transformers is notoriously difficult and laborious; see, e.g., observations given by Liu et al. (2020) as well as scaling laws (Kaplan et al., 2020). In particular, training Transformers requires carefully designed optimizers as well as use of various heuristics. For instance, as illustrated in Figure 1, stochastic gradient descent (SGD)—the workhorse of most deep learning optimization problems—fails to train Transformers effectively. This failure is in contrast to the success of SGD when applied to train convolutional neural networks (CNNs) on vision tasks.

Several recent papers propose a number of different explanations as to why Transformer optimization is so difficult. There is a general consensus in the literature that the loss landscape of Transformers has a number of distinctive features that significantly differ from standard optimization theory assumptions. Most notably, it is empirically verified through various experiments that stochastic gradient noise is heavy-tailed and non-Gaussian (Zhang et al., 2020b; Kunstner et al., 2023) and the loss landscape is significantly ill-conditioned (Zhang et al., 2020a; Jiang et al., 2022; Pan and Li, 2023). In particular, standard assumptions are incapable of dealing with and explaining these observations, and as a result, Transformer optimization has become more of an art than science.

A major obstacle in understanding Transformer optimization is that full-fledged Transformers are extremely complicated to model. One can probe the Transformer's properties by measuring quantities, such as gradient norm or smoothness, but it is much harder to parse the inner-layer workings, and to satisfactorily answer questions such as: *why* does the loss landscape have such features, or *how* do algorithms like Adam perform better than SGD in Transformer training?

Therefore, having an appropriate *mathematical abstraction* is necessary for progress in understanding Transformer optimization—an abstraction that is as simple as possible, while still being able to capture the essence of Transformer optimization. The main message of this paper is that distinctive features of Transformer training also arise in a far simpler setting: the *linear attention model*, without

---

[⋆]Equal contribution, alphabetical order.

nonlinear activations and feedforward networks, being precisely the sought abstraction. We verify that training this model on a low-dimensional linear regression task displays all the distinctive features that have been observed on the full Transformer, suggesting that our surprisingly simple model can serve as a testbed for rigorous understanding of Transformer optimization.

**Main contributions.** We summarize our main contributions as follows:

- We propose the problem of *training shallow linear Transformer model on random linear regression* as a model for understanding Transformer optimization. We verify that this model reproduces all the optimization features and phenomena that have been previously reported for full Transformers.
- We leverage the simplicity of our model to look deeper into how these features arise, by changing settings (e.g., data distribution, the number of layers). Our results reveal that the unique features from previous work get more pronounced in our linear Transformer setting when the data distribution becomes more heavy-tailed, or the number of layers increases.

We expect that such a simple abstraction has great value not only for theoretical research but also for development of optimization methods for Transformers. However, these directions are out-of-scope of this work, and left for future work. As a preliminary, we first survey the previous works that seek to characterize and understand the Transformer optimization landscape.

## 2 DISTINCTIVE FEATURES OF TRANSFORMER OPTIMIZATION

Numerous recent papers have identified a number of distinctive features of the Transformer optimization problem, which set it apart from commonly studied optimization objectives, or even other neural networks such as CNNs. As shown in Figure 1, one of the most striking features is the following:

> Adaptive methods like **Adam are significantly better than SGD**!    (Adam>SGD)

This is in stark contrast with the training of other neural networks (e.g., convolutional neural networks) for which several works have shown that the values of adaptive methods are marginal (Wilson et al., 2017). This phenomenon sparked the interest of the optimization community in investigating the main causes, and subsequently, recent works (Zhang et al., 2020b; Kunstner et al., 2023; Jiang et al., 2022; Pan and Li, 2023) have identified various "unique" features of Transformer optimization.

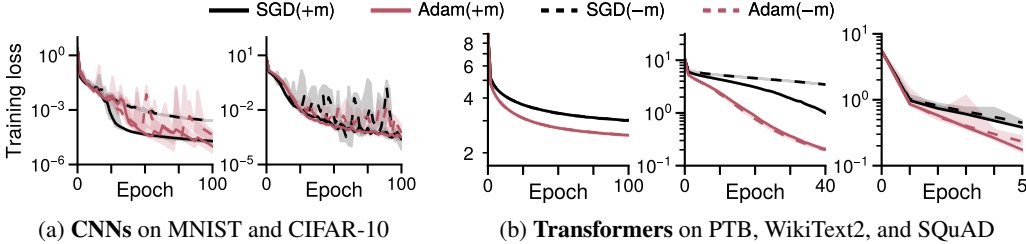

(a) **CNNs** on MNIST and CIFAR-10      (b) **Transformers** on PTB, WikiText2, and SQuAD

Figure 1: Adaptive optimization methods like Adam are much more effective than SGD for training Transformers. This experimental result is taken from (Kunstner et al., 2023, Figure 1). (+m) denotes "with momentum".

In this section, we discuss them one by one in detail, building preliminaries for our main results. In order to discuss each feature, we first give a whirlwind tour on some background in optimization.

### 2.1 A WHIRLWIND TOUR OF (CONVEX) OPTIMIZATION THEORY

For a symmetric matrix $M$, we denote by $\lambda_{\max}(M)$ and $\lambda_{\min}(M)$ the largest and smallest eigenvalue of $M$, and by $\|M\|_2$ the spectral norm of $M$. For simplicity, we assume the training loss function $f$ is twice differentiable. We introduce the following standard concepts in the optimization literature.

- **Lipschitzness.** We say $f$ is $G$-Lipschitz if $\|\nabla f\|_2 \leq G$.
- **Smoothness.** We say $f$ is $L$-smooth if $\|\nabla^2 f\|_2 \leq L$.
- **Strong convexity.** We say $f$ is $\mu$-strongly convex if $\lambda_{\min}(\nabla^2 f) \geq \mu$.
- **Condition number.** The (local) condition number $\kappa_f(x)$ is defined as $\lambda_{\max}(\nabla^2 f(x))/\lambda_{\min}(\nabla^2 f(x))$, provided that $\lambda_{\min}(\nabla^2 f(x)) > 0$.
- **Bounded stochastic gradient noise.** In most SGD analyses, it is assumed that the stochastic gradient $g(x)$ satisfies the *bounded variance* property: $\mathbb{E} \|g(x) - \nabla f(x)\|^2 \leq \sigma^2$.

**Transformers (in practice)**    **Shallow linear Transformers**
(see Subsection 3.1 and Table 1)

**1. Gap between Adam vs. SGD** (Zhang et al., 2020b; Kunstner et al., 2023; Jiang et al., 2022; Pan and Li, 2023):

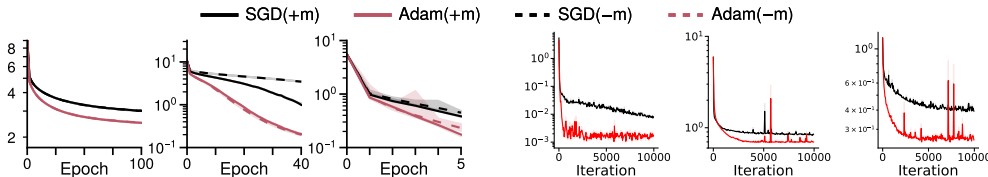

Figure 2: For Transformer optimization, adaptive methods like Adam are strictly better than SGD. (+m) denotes "with momentum" and (-m) denotes without momentum. Our plots only show the momentum variants of SGD and Adam as they perform better in all cases.
Left 3 plots: Full Transformers, from (Kunstner et al., 2023, Figure 1).
Right 3 plots: Shallow linear Transformers (see Settings 1, 2, and 3 from Table 1).

**2. Heavy-tailed stochastic gradient noise** (Zhang et al., 2020b; Kunstner et al., 2023):

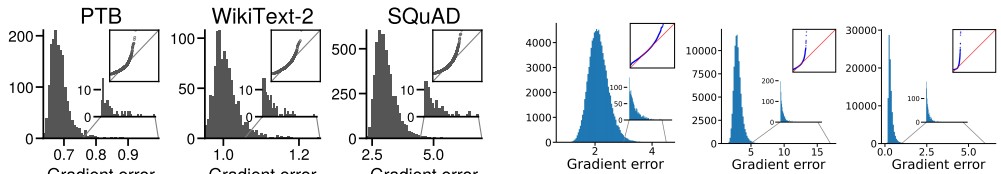

Figure 3: The stochastic gradient noise is heavy-tailed for Transformer optimization. The top-right corner of each plot is the quantile-quantile (q-q) plot between the histogram ($y$-axis) and its best fit Gaussian ($x$-axis). The q-q plot is above the $y = x$ line toward the right, showing its heavy-tailedness.
Left 3 plots: Full Transformers, from (Kunstner et al., 2023, Figure 1).
Right 3 plots: Shallow linear Transformers (see Settings 1, 2, and 3 from Table 1).

**3. Robust condition number of the landscape** (Jiang et al., 2022):

| Layer# | Iteration 750 $R^{SGD}_{med}/R^{Adam}_{med}$ | Iteration 1250 $R^{SGD}_{med}/R^{Adam}_{med}$ | | Iteration 750 $R^{SGD}_{med}/R^{Adam}_{med}$ | Iteration 1250 $R^{SGD}_{med}/R^{Adam}_{med}$ |
|---|---|---|---|---|---|
| 15 | 1.65 (0.65) | 2.01 (1.00) | Setting 1 | 1.76 (0.40) | 1.58 (0.41) |
| 17 | 1.91 (0.53) | 1.43 (0.63) | Setting 2 | 3.14 (0.97) | 5.98 (2.86) |
| 22 | 3.54 (1.21) | 2.28 (1.18) | Setting 3 | 9.57 (13.3) | 6.53 (3.55) |

Figure 4: The comparison of the robust condition number (see Subsection 2.3) between SGD and Adam for Transformer optimization. Numbers in parentheses show standard deviation. Left table: Full Transformers, from (Jiang et al., 2022, Table 1). Right table: Shallow linear Transformers, see Table 1.

**4. Directional smoothness gap between SGD v.s Adam** (Zhang et al., 2020a; Pan and Li, 2023):

Figure 5: log(directional smoothness) against iteration (see Subsection 2.4) for shallow linear Transformers (see Settings 1, 2, 3 from Table 1).

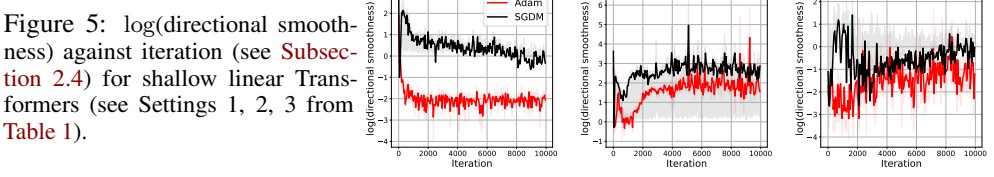

The concepts defined above are typically of great importance in the theory of convex optimization, as the convergence rate of gradient-based optimizers (e.g., gradient descent) typically depend on these quantities. For instance, the convergence rate of gradient descent gets better as the Lipschitzness or smoothness constant gets smaller, or the condition number gets smaller (Bubeck, 2015; Nesterov

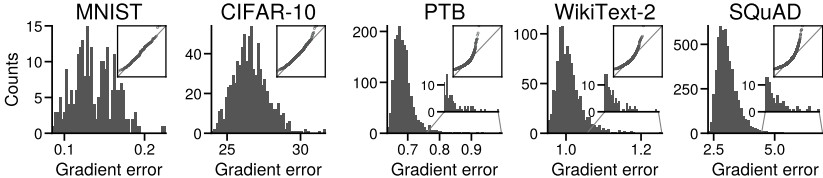

Figure 6: **The heavy-tail stochastic gradient noise for Transformers.** Under the same setting as Figure 1, Kunstner et al. (2023) plot the stochastic gradient noise at the initialization. The top-right corner of each plot is the quantile-quantile (q-q) plot between the histogram ($y$-axis) and its best fit Gaussian ($x$-axis). Notice that the stochastic gradient noise for the convolutional neural networks on vision tasks (MNIST, CIFAR-10) is much less heavy-tailed than the Transformers on NLP tasks. We will revisit this plot in Figure 10.

et al., 2018). Building on these concepts, we now discuss the previous studies on Transformer optimization. Several recent works have connected the difficulties of training Transformers to the unconventional features arising from the loss landscape of Transformer optimization.

## 2.2 HEAVY-TAILED GRADIENT NOISE (ZHANG ET AL., 2020B; KUNSTNER ET AL., 2023)

In (Zhang et al., 2020b) (entitled *Why are adaptive methods good for attention models?*), it was observed that the stochastic gradient is typically more heavy-tailed for Transformer optimization than other neural network optimization. In particular, they make a case that this is opposed to the standard bounded variance condition for SGD analysis – see Figure 3 and Figure 6. They posit that this phenomenon might be one of the main reasons behind the phenomenon (Adam>SGD); they also theoretically show that adaptive step sizes in the form of gradient clipping is required for convergence.

A noteworthy follow-up work by Kunstner et al. (2023) reveal that the heavy-tailed stochastic noise may not explain the full picture. In particular, they compare the full-batch versions (hence no stochastic noise), and notice the phenomenon (Adam>SGD) still hold. Since there is no stochastic noise in this setting, the explanation based on heavy-tailed noise does not apply here.

## 2.3 ILL-CONDITIONED LANDSCAPE (JIANG ET AL., 2022)

In another inspiring work (Jiang et al., 2022), the authors seek to understand the difficulty of Transformer optimization through the lens of condition numbers. In particular, they consider a "robust" condition number defined as $R_{\mathrm{med}}^{\mathrm{OPT}} := \lambda_{\max}(\nabla^2 f)/\lambda_{\mathrm{median}}(\nabla^2 f)$[1], and here the reason for $\lambda_{\mathrm{median}}$ instead of $\lambda_{\min}$ is handle degenerate Hessians. They observe that during Transformer optimization, non-adaptive optimizers like SGD tend to have larger robust condition number than adaptive optimizers like Adam; they posit that this phenomenon is one of the main reasons for (Adam>SGD) – see Figure 4. Jiang et al. (2022) also report that this gap is not there when training convolutational neural networks on image classification tasks, and suggest that this phenomenon may be rooted in unique features of the Transformer which are missing in other popular neural networks.

## 2.4 DIRECTIONAL SMOOTHNESS (PAN AND LI, 2023)

In a follow up work by Pan and Li (2023) (entitled *Toward understanding why Adam converges faster than SGD for Transformers*), the authors again corroborate (Adam>SGD). In addition, they further observe in (Pan and Li, 2023, Figure 6) that proper gradient clipping techniques further accelerate optimization. In order to understand this phenomenon, they propose an explanation based on "directional smoothnesss" along the iterates $x_t$. More formally, they consider the following Taylor expansion along the iterates: for $\eta := \|x_{t+1} - x_t\|$,

$$f(x_{t+1}) - f(x_t) = \nabla f(x_t)^\top (x_{t+1} - x_t) + \frac{1}{2}(x_{t+1} - x_t)^\top \nabla^2 f(x_t)(x_{t+1} - x_t) + O(\eta^3),$$

and define the directional smoothness as $(x_{t+1}-x_t)^\top \nabla^2 f(x_t)(x_{t+1}-x_t)/\|x_{t+1}-x_t\|^2$. In particular, based on the above calculations, one can infer that smaller directional smoothness implies better optimization

---

[1]In fact, in their paper, they instead consider the maximum diagonal entry of the Hessian divided by the median diagonal entry as an approximation of this quantity.

as $f(x_{t+1}) - f(x_t)$ becomes smaller. They claim that the directional smoothness holds the key to understanding (Adam>SGD) (as well as Transformer optimization in general). They also verify that adaptive optimizers tend to have smaller directional smoothness values, and employing gradient clipping further reduces the directional smoothness. Once again, Pan and Li (2023) hypothesize that this feature is unique to Transformers, as they observe that adaptive algorithms can demonstrate *worse directional smoothness* than SGD for, e.g., ResNet training.

## 2.5 GENERALIZED SMOOTHNESS (ZHANG ET AL., 2020A)

We discuss one more noteworthy work (Zhang et al., 2020a) that identifies another unconventional feature. We note that the main motivation of (Zhang et al., 2020a) was not about understanding (Adam>SGD), they also observe their proposed feature in some other non-Transformer networks such as ResNets. The main observation made by (Zhang et al., 2020a) is that the standard smoothness assumption is not suitable for neural network training. Instead, they observe that the spectral norm of Hessian typically grows with the norm of gradient at the current iterate (see Figure 16). Based on this observation, the authors define the following generalized smoothness:

**Definition 1.** *We say $f$ is $(L_0, L_1)$-smooth if $\left\|\nabla^2 f(x)\right\| \leq L_0 + L_1 \left\|\nabla f(x)\right\|$. When $L_1 = 0$, this condition recovers the standard smoothness condition.*

A coordinate-wise version of Definition 1 was considered in (Crawshaw et al., 2022). Under Definition 1, they demonstrate that non-adaptive SGD needs more iterations to converge than an adaptive method based on the global clipping of gradients.

Thus far, we have seen several features identified in the previous works that set Transformer optimization apart from other neural network optimizations. In the next section, we propose a simple yet canonical Transformer model that exhibits all these features.

## 3 LINEAR SHALLOW TRANSFORMERS HAVE THE SAME LOSS LANDSCAPE AS PRACTICAL DEEP TRANSFORMERS

In this section, we show that a simple yet canonical Transformer model exhibits all the features in Section 2. Specifically, the optimization problem to be solved is the training of **linear Transformers on random instances of linear regression**, a model recently proposed for understanding of in-context learning (Garg et al., 2022; Akyürek et al., 2022; von Oswald et al., 2023; Ahn et al., 2023b; Zhang et al., 2023; Mahankali et al., 2023).

### 3.1 LINEAR TRANSFORMER ON LINEAR REGRESSION

**Data distribution.** The data distribution can be thought of as the random instances of linear regression. Concretely, for $i = 1, 2 \ldots, n+1$, let $x^{(i)} \in \mathbb{R}^d$ be drawn *i.i.d.* from a distribution $D_{\mathcal{X}}$. We then draw $w_\star \sim D_{\mathcal{W}}$ and then generate the scalar responses $y = [\langle x^{(1)}, w_\star \rangle, \ldots, \langle x^{(n)}, w_\star \rangle] \in \mathbb{R}^n$. Now the input of the data set consists of these linear regression examples:

$$\text{Input matrix: } Z_0 = \begin{bmatrix} x^{(1)} & x^{(2)} & \cdots & x^{(n)} & x^{(n+1)} \\ y^{(1)} & y^{(2)} & \cdots & y^{(n)} & 0 \end{bmatrix} \in \mathbb{R}^{(d+1) \times (n+1)}.$$

The goal is to predict the missing $y^{(n+1)}$, as we detail below.

**Optimization objective.** Let $\mathsf{TF}_L(\cdot; W) : \mathbb{R}^{(d+1) \times (n+1)} \to \mathbb{R}$ denote the prediction of the $L$-layered linear Transformer with parameters $W$. Our optimization objective is given by

$$f(W) := \mathbb{E}_{(Z_0, w_\star)} \left[ \left( \mathsf{TF}_L(Z_0; W) - w_\star^\top x^{(n+1)} \right)^2 \right].$$

In words, we train the linear Transformer to predict $y^{(n+1)}$ using $\mathsf{TF}_L(Z_0; W)$; we will formally define the linear Transformer architecture below. This objective was the center of study in a number of recent empirical and theoretical works on understanding Transformers (von Oswald et al., 2023; Ahn et al., 2023b; Zhang et al., 2023; Mahankali et al., 2023).

**Linear Transformer (self-attention) architecture.** We will now present the neural network architecture that will be used throughout this paper. Given matrices $P, Q \in \mathbb{R}^{(d+1)\times(d+1)}$, we define the **linear self-attention** architecture as

$$\mathsf{Attn}_{P,Q}(Z) = PZM(Z^\top QZ) \quad \text{where} \quad M := \begin{bmatrix} I_n & 0 \\ 0 & 0 \end{bmatrix} \in \mathbb{R}^{(n+1)\times(n+1)}. \tag{1}$$

Finally, for a positive integer $L$, we define an $L$-**layer linear Transformer** $\mathsf{TF}_L$ as a stack of $L$ linear attention units. Specifically, let the output of the $L^{\text{th}}$ layer attention, $Z_L$, be recursively defined as

$$Z_{\ell+1} = Z_\ell + \frac{1}{n}\mathsf{Attn}_{P_\ell, Q_\ell}(Z_\ell) \quad \text{for } \ell = 0, 1, \dots, L-1.$$

Then we define $\mathsf{TF}_L(Z_0; \{P_\ell, Q_\ell\}_{\ell=0}^{L-1}) = -[Z_L]_{(d+1),(n+1)}$, i.e., the $(d+1, n+1)$-th entry of $Z_L$. The reason for the minus sign is to be consistent with (von Oswald et al., 2023; Ahn et al., 2023b), where such a choice was motivated by theoretical considerations.

We emphasize here that the linear attention unit, defined in (1), differs from the standard attention unit in (Vaswani et al., 2017): our architecture does not have feedforward networks, and we use a single matrix $Q$ to represent the product of key, query matrices. More importantly, *we remove the softmax activation outside $Z^\top QZ$.* There are two key reasons for our choice:

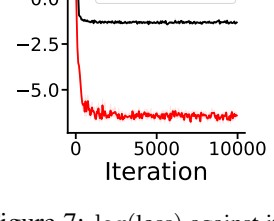

1. The linear attention unit is *much better suited to the task of linear regression.* For instance, (von Oswald et al., 2023, Appendix A.9) demonstrates that the performance of softmax Transformer with twice many heads matches that of linear Transformers; in other words, we need two softmax attention heads to recover the performance of a single linear head. In Figure 7, we show that linear attention performs significantly better than standard attention with softmax.

Figure 7: $\log(\text{loss})$ against iteration. Comparison between linear attention and softmax attention for the 3-layer Transformers. Note that the loss of linear Transformer decreases much faster.

2. Our goal in this paper is to *find the simplest abstraction* which is representative of the Transformer's optimization landscape. As we will see in Subsection 3.2, the loss landscape of the linear Transformer well approximates that of the actual Transformer, even without the softmax activation, feedforward networks, and other components of standard Transformers.

We also note that the key-query matrix is parametrized by a single matrix $Q$, which is another difference relative to standard Transformers. We make such a parametrization for simplicity, and in the left plot of Figure 8, we verify that the loss plot for the standard parametrization is largely similar to ours. We also remark that the lack of softmax may result in different learned attention scores. In particular, it may lead to denser attention scores than the attention scores for softmax Transformers (Oymak et al., 2023; Li et al., 2023a;b). On the other hand, the sparsity of learned attention scores depends on the data distribution; for instance, we observe that orthogonal covariates (as in (Huang et al., 2023)) lead to sparser attention scores for both linear and softmax Transformers.

## 3.2 LINEAR TRANSFORMERS AS A FRUITFUL ABSTRACTION

| $(d = 5)$ | **Setting 1** (Ahn et al., 2023b) | **Setting 2** (fewer covariates) | **Setting 3** (heavy-tailed covariates) |
|---|---|---|---|
| #contexts $n$ | 20 | 5 | 20 |
| distribution of $x^{(i)}$ | $\mathcal{N}(0, I_d)$ | $\mathcal{N}(0, I_d)$ | $\sqrt{\Gamma_{0.1,10}} \cdot \mathrm{Unif}(\mathbb{S}^{d-1})$ |
| distribution of $w_\star$ | $\mathcal{N}(0, I_d)$ | $\mathcal{N}(0, I_d)$ | $\mathcal{N}(0, I_d)$ |

Table 1: Settings for (the right-side plots of) Figures 2, 3, 4, 5, and 16.

**Setting for the experiments.** Having established the framework in Subsection 3.1, we now describe details of our experiments. Our base-setup is the 3-layer linear Transformer, with 5-dimensional covariates, i.e. $(L = 3, d = 5)$. This is the minimally complex setting that still recovers all of the

discussed features of full Transformers. Transformers with larger $L$ or $d$ are qualitatively similar to the $(L = 3, d = 5)$ setting, and we provide such an example in the right plot of Figure 8.

Our "default" setup is Setting 1 of Table 1, where the context consists of 20 context demonstrations; each context covariate is sampled from the standard Gaussian, i.e., $x^{(i)} \sim \mathcal{N}(0, I_d)$, and we draw $w_\star \sim \mathcal{N}(0, I_d)$. This is consistent with previous works (Garg et al., 2022; Akyürek et al., 2022; von Oswald et al., 2023; Ahn et al., 2023b). In order to see the effect of nonlinearity in data distribution, we conduct an additional set of experiments for a **nonlinear regression** where the covariates are distorted by a multilayer perceptron (MLP) with nonlinear activations; see Appendix B for details.

In order to understand the effect of context length, we also consider the setting when context length $n = 5$ instead; this is Setting 2 of Table 1.

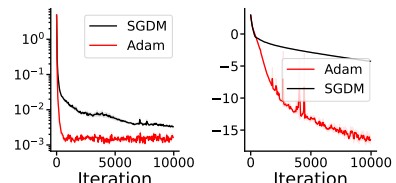

Finally, to investigate the effect of heavy-tailed covariates on various aspects of the loss landscape, we consider Setting 3 in Table 1, where we draw each $x_i$ instead uniformly from the unit sphere, and then scale it by the square root of a heavy-tailed Gamma random variable with shape parameter $k = 0.1$ and scale parameter $\theta = 10$. Furthermore, in Subsection 4.1, we study the effect of heavy-tailedness of the covariates in more detail.

Figure 8: **Left:** The case when Transformer is parameterized by separate $Q, K$ (query, key) matrices, instead of a single matrix as in (1). The setting is the same as Setting 1 in Table 1. **Right:** The setting of 8-layer linear Transformer with covariate dimension $d = 20$ and context length $n = 60$.

For each different setting, we pick the best learning rate from a grid search over 10 different choices. We choose the momentum parameter 0.9 for SGD, and $\beta_1 = \beta_2 = 0.9$ for Adam. We also employ the (global) gradient clipping where the thresholds are chosen to be 1 for all settings (i.e., the clipped gradient direction is the same as the non-clipped direction). All the experiments are run over 6 different random seeds. See Appendix A for details.

**Discussion of results.** Below we provide detailed discussion of the results.

1. **Gap between SGD and Adam.** In Figure 2 (right), we plot the training loss for the three settings in Table 1. Notice that we observe the phenomenon (Adam>SGD) over three different settings, to different extents. These loss behaviors resemble those of the practical Transformer optimization (left plots of Figure 2).
2. **Heavy-tailed stochastic noise.** In Figure 3 (right), following (Zhang et al., 2020b; Kunstner et al., 2023), we plot the stochastic gradient noise at the initialization. Notice the similarity between the left plots and the right plots, showing that the shallow linear Transformers also exhibit the heavy-tailed stochastic gradient noise phenomenon.
3. **Condition number of the landscape.** Following (Jiang et al., 2022), we measure the "robust" condition numbers of different optimizers along the trajectory. Figure 4 shows that the condition numbers of adaptive methods are lower than those of SGD, similar to (Jiang et al., 2022).
4. **Directional smoothness.** As observed by previous works (Zhang et al., 2020a;b; Pan and Li, 2023), in our experiments, we also observe that Adam has better directional smoothness than SGD, which correlates with the speed-up of Adam over SGD. We present this in Figure 5.
5. **Generalized smoothness.** As discussed in Subsection 2.5, the generalized smoothness condition of Zhang et al. (2020a) might not be a unique feature to Transformer optimization. Nevertheless, interestingly, we also observe such a phenomenon (to a certain extent) in shallow linear Transformer optimization as shown in the right plots of Figure 16.

In this section, we have seen that simple linear Transformers described in Subsection 3.1 suffice to recover all the main features identified in previous works (Section 2). In the next section, we take advantage of the concreteness and simplicity of our linear Transformer to explore and understand the role of heavy-tailedness in data distribution and depth of the network.

## 4 UNDERSTANDING FEATURES BASED ON LINEAR TRANSFORMERS

The main advantage of our toy linear Transformer comes from its simplicity and concreteness. In particular, thanks to the concreteness of the setting, one can conduct various "controlled" experiments to understand the features observed in Subsection 3.2. Recall that the data set used in our experiments

consists of nothing but random linear regression instances. This data set is far simpler and more concrete than the language modeling data sets (e.g., Wikipedia texts, question&answering) of the previous works discussed in Section 2.

We first take advantage of the concreteness of our data distribution, and look deeper into how the main distinctive features of Transformer optimization arise. We first investigate how the "heavy-tailedness" of the data distribution affects the extent of the features from Section 2.

## 4.1 EFFECT OF DATA DISTRIBUTION

Given that we observe the "heavy-tailedness" of stochastic gradient noise, perhaps a natural question to ask is the following:

***Q. Does the "heavy-tailedness" of data distribution exacerbate the features in Section 2?***

**Settings.** In order to investigate the above question, we consider the following distributions for the covariates $x^{(i)}$'s of linear regression for $(L = 3, d = 5, N = 20)$:

**- Spherical covariates.** We sample $x^{(i)}$'s uniformly at random from the unit sphere $\mathbb{S}^{d-1}$.

**- Heavy-tailed covariates.** We first sample $x^{(i)}$'s uniformly at random from the unit sphere $\mathbb{S}^{d-1}$, and then multiply each covariate by a random scale drawn *i.i.d* from a heavy-tailed distribution, specifically the square root of a Gamma random variable from $\Gamma_{k,\theta}$. Note that $k = 2.5$ and $\theta = 2$ precisely corresponds to the case where $x^{(i)} \sim \mathcal{N}(0, I_5)$. In our experiments, we use $k = 0.1$ and $\theta = 10$ to make the distribution more heavy-tailed, while keeping the variance the same.

**Discussion.** We now discuss the experimental results one by one:

▶ In Figure 10, we see that "heavy-tailed"-ness of covariates is reflected in the "heavy-tailed"-ness of the stochastic gradient. Notably, the contrast between the two plots in Figure 10 reminds us of the contrast we see between CNNs and Transformers in Figure 6.

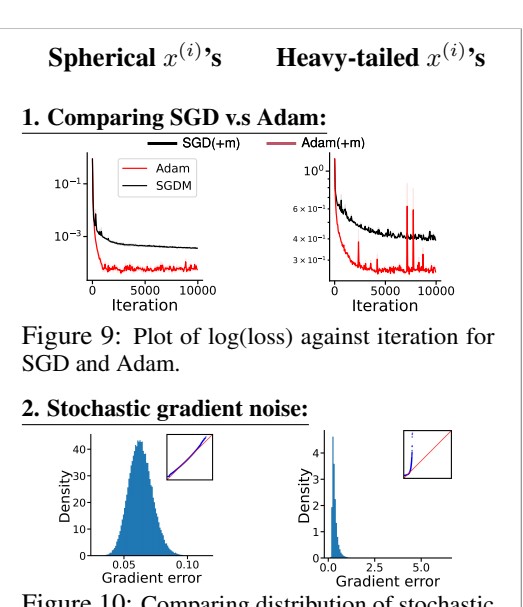

**Spherical $x^{(i)}$'s      Heavy-tailed $x^{(i)}$'s**

**1. Comparing SGD v.s Adam:**

Figure 9: Plot of log(loss) against iteration for SGD and Adam.

**2. Stochastic gradient noise:**

Figure 10: Comparing distribution of stochastic gradient noise at the initialization

**3. Robust condition number:**

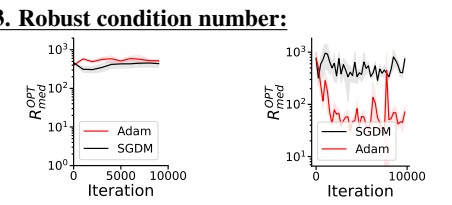

Figure 11: Comparing the robust condition number from Jiang et al. (2022)

▶ In Figure 11, it appears that there is some correlation between the gap in robust condition number, and the "heavy-tailed"-ness of the data distribution, with heavier tails leading to larger gaps.

▶ Finally, Figure 9 shows how the optimization speed of SGD and Adam vary with the heavy-tailedness of covariates. First, given spherical (light-tailed) covariates, both SGD and Adam converge much faster than Gamma-scaled (heavy-tailed) covariates. On the other hand, the *relative gap* between the speed of Adam and SGD does not seem to improve noticeably under light-tailed noise.

▶ Together, Figure 9 and Figure 10 show that the relationship between heavy-tailed gradient noise and optimization speed may be a little more complicated than suggested in (Zhang et al., 2020b). Specifically, adaptivity seems to be equally beneficial regardless of the heavy-tailedness of the gradient noise. Instead, these two plots seem to align more with the message in (Kunstner et al., 2023) – that noise may not be the sole contributor of (Adam>SGD).

We next take advantage of the concreteness of our model, and investigate the effect of the number of layers on the optimization.

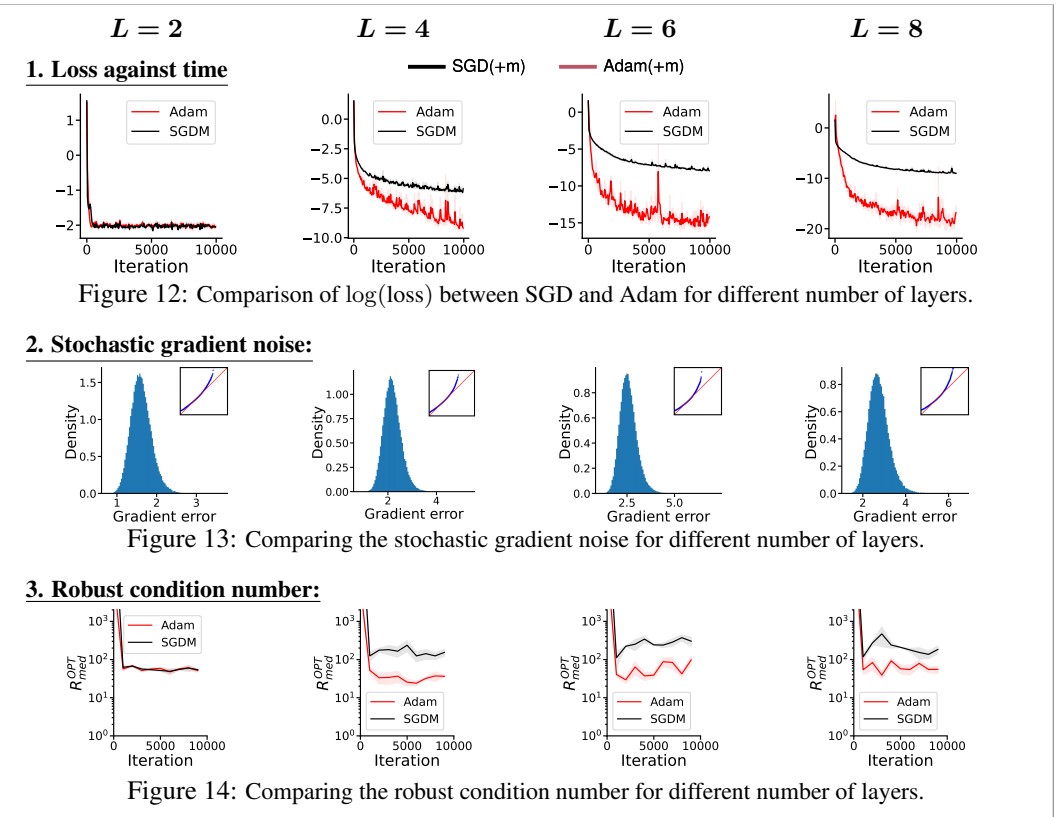

Figure 12: Comparison of $\log(\text{loss})$ between SGD and Adam for different number of layers.

Figure 13: Comparing the stochastic gradient noise for different number of layers.

Figure 14: Comparing the robust condition number for different number of layers.

## 4.2 EFFECT OF MORE LAYERS

We investigate the effect of the number of layers $L$ on the optimization. Specifically,

**Q.** *Will a deeper linear Transformer exacerbate the features in Section 2?*

**Settings.** In order to investigate the above question, we consider repeating the experiments in Setting 1 of Table 1 for the number of layers $L \in \{2, 4, 6, 8\}$.

**Discussion.** We present the experimental results one by one:

▶ As one can see from Figure 12, the gap in loss between adaptive methods and SGD become more and more pronounced as we increase the number of layers.
▶ On the other hand, the absolute value of the loss decreases with increasing depth, for both SGD and Adam, which makes sense considering the larger capacity of deeper models.
▶ In Figure 13, we see that the stochastic gradient noise for the case of $L = 6, 8$ are more heavy-tailed than the case of $L = 2, 4$.
▶ Lastly, we observe in Figure 14 that the gap in the robust condition number of SGD and Adam is more pronounced in deeper models ($L = 4, 6, 8$) than the shallow model ($L = 2$).

## 5 CONCLUSION

The complexity of modern neural networks, especially Transformers, often eludes precise mathematical understanding, and hence calls for such "physics-style" approaches (c.f. Zhang et al. (2022); Ahn et al. (2023a); Abernethy et al. (2023); Allen-Zhu and Li (2023); Li et al. (2023b); Dai et al. (2023)) based on simplified models. This work presents a concrete addition to this viewpoint, and it builds a valuable, realistic proxy for understanding Transformers. However, our findings currently lack a solid theoretical foundation, and our linear regression setting may not fully capture the features of the language data utilized in Transformer optimization. We hope that our work will serve as the stepping stone for building a more precise theory of Transformer optimization, as well as contributing to the development of efficient training methods for Transformers.

## ACKNOWLEDGEMENTS

This work stems from a group project at MIT; we thank the collaborators in the group, Hadi Daneshmand, Haochuan Li, Zakaria Mhammedi, Swati Padmanabhan, Amirhossein Reisizadeh, and William Wang for their time and intriguing discussions.

Kwangjun Ahn and Ali Jadbabaie were supported by the ONR grant (N00014-23-1-2299) and MIT-IBM Watson as well as a Vannevar Bush fellowship from Office of the Secretary of Defense. Xiang Cheng and Suvrit Sra acknowledge support from NSF CCF-2112665 (TILOS AI Research Institute) and an NSF CAREER award (1846088). Minhak Song and Chulhee Yun were supported by Institute of Information & communications Technology Planning & Evaluation (IITP) grant (No. 2019-0-00075, Artificial Intelligence Graduate School Program (KAIST)) funded by the Korea government (MSIT), two National Research Foundation of Korea (NRF) grants (No. NRF-2019R1A5A1028324, RS-2023-00211352) funded by the Korea government (MSIT), and a grant funded by Samsung Electronics Co., Ltd.

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

# Appendix

## A    HYPERPARAMETERS FOR THE EXPERIMENTS

In this section, we summarize the choice of hyperparameters for Subsection 3.2 and Section 4. We choose the momentum parameter $0.9$ for SGD, and $\beta_1 = \beta_2 = 0.9$ for Adam. We also employ the (global) gradient clipping where the thresholds are chosen to be $1$ for all settings (i.e., the clipped gradient direction is the same as the non-clipped direction). The choice of learning rates is summarized in the following table for (1) Setting 1 from Table 1, (2) Setting 2 from Table 1, (3) Setting 3 from Table 1, (4) Spherical covariates setting of Subsection 4.1, (5) Heavy-tailed covariates setting of Subsection 4.1, (6) $L = 2$ setting of Subsection 4.2, (7) $L = 4$ setting of Subsection 4.2, (8) $L = 6$ setting of Subsection 4.2, and (9) $L = 8$ setting of Subsection 4.2

| lrs of | (1) | (2) | (3) | (4) | (5) | (6) | (7) | (8) | (9) |
|--------|-----|-----|-----|-----|-----|-----|-----|-----|-----|
| SGDM | 0.02 | 0.01 | 0.02 | 5 | 0.02 | 0.1 | 0.05 | 0.05 | 0.05 |
| Adam | 0.005 | 0.02 | 0.02 | 0.1 | 0.02 | 0.1 | 0.05 | 0.05 | 0.02 |

Table 2: The choice of learning rates for experiments.

## B    ADDITIONAL EXPERIMENTS FOR NONLINEAR REGRESSION

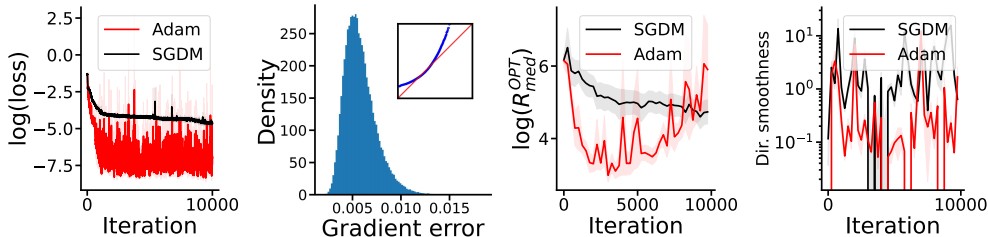

Figure 15: The results for the nonlinear regression where the covariates are distorted by a ReLU network.

In this section, we consider the case of nonlinear regression, where the covariates $x^{(i)}$'s of the linear regression are distorted by a multilayer perceptron (MLP). Let us describe the setting:

- Analogous to the Setting 1 of Table 1, *i.e.*, $N = 20$, $d = 5$, $x^{(i)} \sim \mathcal{N}(0, I_d)$, and $w_\star \sim \mathcal{N}(0, I_d)$.
- On the other hand, to generate the responses $y^{(i)}$, we first fix a randomly generated one-hidden-layer multilayer perceptron (MLP) with ReLU activation that we denote by $\mathsf{MLP} : \mathbb{R}^5 \to \mathbb{R}^5$ with 5 hidden neurons and consider $y^{(i)} = \langle w_\star, \mathsf{MLP}(x^{(i)}) \rangle$. In particular, we use the code `nn.Sequential(nn.Linear(5, 5),nn.ReLU(),nn.Linear(5, 5))` (where `nn` is the `torch.nn` in PyTorch) for generating the random ReLU network MLP.
- In order to cope with the MLP, in our linear Transformer architecture, we add an additional ReLU MLP layer with 15 hidden neurons before the linear Transformer blocks.

For the choice of learning rates, the optimal learning rates for this setting is 0.01 for Adam and 0.05 for SGD. As one can see from Figure 15, we get similar plots to the case of linear regression.

## C   ADDITIONAL PLOTS

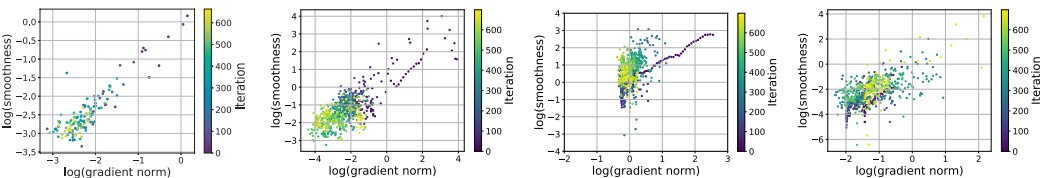

Figure 16: The plot of $\log(\|\nabla f(x_t)\|)$ against $\log$(smoothness). Following (Zhang et al., 2020a), we measure the directional smoothness instead of $\|\nabla^2 f(x_t)\|_2$. We observe similar trends with $\|\nabla^2 f(x_t)\|_2$.
Left plot: LSTM from (Zhang et al., 2020a, Figure 1). Right 3 plots: Shallow linear Transformers trained with Adam, see Settings 1, 2, 3 in Table 1.

