# OpenReview forum: "Linear attention is (maybe) all you need (to understand Transformer optimization)"
_ICLR.cc/2024/Conference — ICLR 2024 poster_

### Official Review · Reviewer_gJVN · 2023-10-22

**Soundness:** 3 good
**Presentation:** 3 good
**Contribution:** 3 good
**Rating:** 6
**Confidence:** 4

**Summary:**

This paper aims at deriving a simplified model for theoretical study of Transformers' optimization. The authors study a pure-attention without softmax Transformer variant (which is called linear attention) on a synthetic regression task. By conducting extensive empirical study, the authors claim that the proposed linear attention Transformer exhibits similar optimization properties that have been observed in the training of vanilla Transformers. Thus, this paper concludes that the proposed linear attention Transformer can be a realistic proxy for understanding Transformer optimization.

**Strengths:**

- The paper is well-written and easy to follow.
- Sec. 2 offers a concise and informative overview of existing study on Transformers' optimization. It's helpful for readers to digest the context of this study.
- The empirical study and comparisons look solid and convincing. I'm looking forward to optimization theories based on the proposed simplified model.

**Weaknesses:**

- The proposed simplified model does not have feed forward network (FFN), which is yet another difference from standard Transformers. The authors should explicitly point this out (e.g., in the introduction and the formulation of model architecture) in the paper.

- I strongly recommend the authors to summarize the detailed training recipes (e.g., hyperparameters) in a table and release the code for calculating relevant quantities (e.g., gradient error, robust condition numbers) to facilitate reproducibility and follow-up research.

- In the beginning of Sec. 3.1, $x^{(i)}$ is defined as a row vector. But in the input matrix it seems that one has to interpret $x^{(i)}$ as a column matrix. Moreover, the matrix shape here is very confusing. The input of $F$ is $(n+1)\times (d+1)$, while $Z_0$ is $(d+1)\times (n+1)$. I suggest following the "_Attention Is All You Need_" paper and using shape $(n+1)\times (d+1)$ consistently. The authors should fix the notation and formulas here.

**Minor issues.**

- Sec. 2.1: _The features defined above are typically of ..._ $\to$ _The concepts defined above are typically of ..._

- I recommend the authors to capitalize the first letter of "Transformers".

**Questions:**

The question here is just for clarification.

- What is the box in the upper right of the left 3 plots in Fig. 3?

---

> ### Author Response · Authors · 2023-11-14
> **Thank you for your comments!**
>
> Thank you for your fruitful comments! We revised our paper based on your comments and all the edits are highlighted in orange.
>
> - We added in our revision that "our setting doesn't have FFN" in the places that you suggested.
>
> - For the dimensions in the beginning of Section 3.1, those were typos. Thank you for catching them. We now corrected them.
>
> - We also corrected the "minor issues" raised by you; we also capitalize "Transformers" throughout the manuscript.
>
> - For the upper right box in  Fig. 3 are the quantile-quantile plots (QQ-plots) to see how heavy-tailed the distributions are. We added the QQ plots for our experiments as well as descriptions in our paper.
>
> - We added the summary of hyperparameter choice in Appendix A. Also, we attached the code in the supplementary material.

---

> > ### Comment · Reviewer_gJVN · 2023-11-14
> >
> > I thank the authors for the timely response. My concerns are mostly addressed. I maintain my positive rating for the paper.

---

### Official Review · Reviewer_dN4V · 2023-10-31

**Soundness:** 3 good
**Presentation:** 3 good
**Contribution:** 3 good
**Rating:** 8
**Confidence:** 3

**Summary:**

This paper offers empirical evidence highlighting parallels between shallow linearized transformer models and the dynamics observed during practical Transformer training. These properties span various well-known aspects: the superiority of adaptive methods over SGD, the presence of heavy-tailed stochastic gradient noise, a robust condition number characterizing the optimization landscape, and the directional smoothness gap. The findings advocate for the potential of simple linear transformers as an insightful and accessible model to elucidate the mystery of Transformer optimization trajectories.

**Strengths:**

The presented work is overall clear and of notable quality. Its innovative aspect is the unexpected empirical evidence indicating that, under certain conditions, shallow linear transformers can closely resemble conventional attention models, revealing a significant similarity between the simple model and its real-world analogs.

Given the historical challenges associated with analyzing attention mechanisms involving softmax or other nonlinearities, it is commendable to identify such marked resemblances under broad observations using a simplified framework. This study illuminates the theoretical analysis for optimization on linear transformers, which makes a valuable contribution to the literature.

**Weaknesses:**

The paper is somewhat limited in its data distribution and optimization objective setting to support its conclusion. The data distribution discussed in this setting focuses on in-context linear regression tasks—a largely statistical setting, rather than one that closely resembles language. This might be attributed to the limited expressivity of linear transformers.

I fully understand that the authors are trying to *find the simplest abstraction* as a representative of the transformer’s landscape. However, since the landscape is likely heavily dependent on the data distribution, it would be more convincing to see a variety of data models that exhibit the same phenomenon in simple models. Furthermore, it's difficult to assert that the linearized model is sufficient to understand Transformer optimization, especially since the properties verified in this paper represent only a portion of the optimization narrative.

**Questions:**

- Are there other possible tasks that can be expressed by linear transformers? Do those tasks on simple linear transformers also exhibit those behaviors?
- If we add other linearities (such as softmax) to the shallow neural networks, can you also show those properties along the training trajectory?

---

> ### Author Response · Authors · 2023-11-17
> **Response**
>
> Thank you for appreciating the values of our work and constructive comments! We also believe that having a simplified framework is valuable for the future rigorous study on Transformer optimization.
>
> We also appreciate your comment "**I fully understand that the authors are trying to find the simplest abstraction as a representative of the transformer’s landscape.**"
> That is precisely the main scope of this work.
>
>
> To address your comments about "the landscape being likely dependent on the data distribution," in **Appendix B**, we conducted an additional set of experiments for a nonlinear regression. In particular, we consider the case where the covariates $x^{(i)}$ are distorted by a ReLU network $MLP$, i.e., the responses are generated as $y^{(i)} = \langle w_\star , MLP(x^{(i)}) \rangle$. In order to cope with the nonlinearity we also added a MLP in our Transformer architecture.
> As you can see from **Appendix B**, the results for the nonlinear regression are largely similar to the linear regression. Although we agree that nonlinear regression also can't fully explain the practical language modeling, we hope that our additional set of experiments addresses your concern.

---

### Official Review · Reviewer_bjjH · 2023-11-01

**Soundness:** 3 good
**Presentation:** 3 good
**Contribution:** 3 good
**Rating:** 6
**Confidence:** 3

**Summary:**

This paper identifies "use shallow linear Transformers for regression task" as a simple setup that serves as an effective mathematical abstraction for studying Transformer optimization.
- The linear Transformer has attention (inner product) and residual connection, without softmax and MLP. For attention, a single $Q$ matrix is used to replace the product $W_Q^\top W_K$.
- The aim is to find the _simplest_ setup as a surrogate for studying optimization: this paper uses a 3-layer network with 5-dim covariate, on the linear regression task.

**Strengths:**

- The simple setup is able to reproduce several characteristics of the training dynamics, which are not captured by canonical optimization theory. These include heavy-tailed gradient noise, robust condition number (larger for SGD than for Adam), as well as better directional and generalized smoothness from Adam.
- The synthetic setup allows for more precise control.
    - The paper varies the number of layers, data distribution, and context length.
    - It's affordable to grid search over 10 learning rates.

**Weaknesses:**

- The main claim is that the simplified setup can be used as a proxy for larger scale systems. However, some connections are drawn by having "qualitatively similar" results.
  - For example it's known that higher-order moments of the gradient noises matter, which cannot be captured by examining plots visually.
  - e.g. How to judge whether the noises are similar based on Fig 3, or that 8-layer is qualitatively similar to 3-layer based on Fig 8?
- The current empirical results are for confirming existing findings, but it's unclear what new discoveries can be enabled by this simplified setup.

**Questions:**

- What are some phenomena that are not able to be captured in this simplified setup?
    - e.g. using the $Q$ parameterization rather than $W_Q^\top W_K$ might lead to different implicit regularization effects.
    - There might be effects about architectural choices, e.g. consequences and limitations from the softmax.
    - In general, it would be helpful to better clarify what is consider in-scope and what is not.
- Could you give some examples of new discovery or actionable advice from the simple setup?
- The current setup is entirely linear; if the data is non-linear, would there be similar optimization characteristics?

---

> ### Author Response · Authors · 2023-11-14
> **Thank you for your comments!**
>
> Thank you for your constructive comments. We agree with your comment that our simple setting lets us do more precise controlled experiments.
>
> Let us address your points one by one. We revised our paper based on your comments and all the edits are highlighted in orange.
>
> -- **More quantitative way of checking noise distribution?** Following [Kunstner et al., 2023], we added the quantile-quantile plot (QQ-plot) in our revision. A QQ-plot is a plot of the quantiles of two distributions against each other. In particular, we add QQ-plots that compare the quantiles of the stochastic gradient noise (y-axis) against those of its best-fit Gaussian distribution. One can see that QQ-plots go beyond $y=x$ line toward the right, suggesting that the stochastic noises are indeed heavy-tailed. Also, for the heavy-tailed covariates experiments and the more layers experiments, we now can also see clearly for which cases the noises become more heavy-tailed.
>
> -- **Regarding the $W_Q W_K$ parametrization.** Thank you for your question. In the left plot of Figure 8, we added a result for the standard Transformer parameterization of $W_Q W_K$, and observed that the result looks similar to our simplified parameterization.
>
> -- **Regarding softmax activation.** We mostly considered linear Transformer since it's better suited for the task of solving linear regression in-context as shown in Figure 7.
>
> -- **Clarifying the scope of this work** As per your suggestion, we further clarified the main scope of this work in our introduction, by saying "....development of efficient optimization methods for Transformers. However, such directions are out-of-scope of this work, and left for future work."
>
> -- **Our setting entirely linear?** We would like to clarify that despite removal of nonlinearity, the linear Transformer is still highly nonlinear function due to the attention mechanism. In particular, even the output of single layer linear Transformer is a multivariate 3rd order polynomial function of the input.
>
> **[Updated Response]**: In order to address your concerns about the linearity in the data distribution, in our newest revision, we conducted an additional set of experiments for the nonlinear regression, and it's presented in **Appendix B**.

---

> > ### Comment · Reviewer_bjjH · 2023-11-18
> >
> > Thank you for the clarifications and the additional results! I have a few more questions please:
> >
> > - For Fig 3, the distributions in practice vs synthetic are clearly different based on the qq-plots; could you comment on how much you expect this difference to matter for analyses?
> > - For Fig 15, why does Adam's robust conditional number increase in the end? The reason for me to ask about results on non-linear distribution is that I wonder what the linear setup would fail to capture, and how transferable would the conclusions on the linear setup transfer to more general cases.
> > - Minor comment: please make the y-axis consistent across figures; e.g. the loss scales in Fig 8 left and right, and Fig 15 left.

---

> > > ### Author Response · Authors · 2023-11-19
> > > **Thank you for your questions!**
> > >
> > > Thank you for your follow up questions! Hope our answers below address your questions.
> > >
> > > - For Figure 3, we first would like to emphasize that the qq-plots in practice and synthetic are **both demonstrating heavy-tail distributions**, which is indeed our main claim. We assume that by "clearly different", the reviewer is referring to the fact that the three qq-plots on the right (synthetic setup) diverge much more sharply from the red diagonal line. We highlight that the sharpness of the divergence **depends on the choice of data distribution** -- a heavier-tailed distribution leads to a sharper divergence. For instance, **we could produce qq-plot more similar to the LHS (practical setup) of Figure 3 if we chose Gamma with shape parameter 1** instead of 0.1.
> > >
> > > - For Figure 15, Adam's robust condition number increases near the end, and we attribute this to the property of the training loss landscape near the minimum, as we detail below. In particular, we in fact did observe **this phenomenon for both linear and non-linear settings.** It is not shown in the linear setting because **Figure 4 table only shows the robust condition number up to iteration 1250**, to be consistant with the previous work (LHS of Figure 4). For comparison, **the robust condition number in Figure 15 only increases after iteration 5000**.
> > >
> > >    Now we breifly provide an intution behind why the loss landscape is unstable near the minimum. Borrowing the conclusions from previous works [1,2], the minimizers of the landscape corresponds to implementing an algorithm similar to gradient descent. On the other hand, we empirically observe that the "stepsize" of the learned algorithm is usually near the maximum possible, and slightly increasing the "stepsize" by even a little will cause the Transformer output to diverge. We suspect this observation to be the main cause behind the unstable landscape near the minima. We hope this answer clarifies some of your questions.
> > >
> > > - Thank you for the suggestion on y-axis consistency. We will update these plots accordingly in the next draft.
> > >
> > > [1] [Transformers Learn In-Context by Gradient Descent](https://proceedings.mlr.press/v202/von-oswald23a/von-oswald23a.pdf)
> > >
> > > [2] [Transformers learn to implement preconditioned gradient descent for in-context learning](https://arxiv.org/abs/2306.00297)

---

### Official Review · Reviewer_GNbE · 2023-11-01

**Soundness:** 3 good
**Presentation:** 3 good
**Contribution:** 3 good
**Rating:** 6
**Confidence:** 4

**Summary:**

This paper studies the property of the shallow linear Transformer model. The evaluated phenomenon includes comparing Adam and SGD, heavy-tailed gradient noise, condition number, smoothness, etc. The experimental results show that the linear Transformer model reproduces the phenomena that have been observed for full Transformers. The results also show that more heavy-tailed data distribution and more layers can enhance the conclusion.

--------------------------------------------------------

**After rebuttal**: Thank you for your response. I am sorry for the late reply. I am satisfied with the feedback and revisions. I will keep my rating as 6 and support you. A minor point and suggestion: Please clarify what the four figures in Fig. 15 are to be compared within the main body. Later, I realize you are mentioning some figures on page 3. It is not a big issue anyway.

**Strengths:**

This work is novel in terms of new experiments on linear transformers from many aspects of evaluation. The paper is well-written and clear to follow. The studied problem is interesting to the community, from my understanding. The designed experiments are concise to support the conclusion.

**Weaknesses:**

1. This work lacks theoretical understanding or explanation after the experiments in Section 3.2, 4.1, 4.2.
2. The limitation is not discussed. One thing that should be emphasized in the introduction or the abstract is that the experiments are linear regression, which is a good fit for linear Transformers.

**Questions:**

1. One conclusion may not be obvious for linear Transformers compared with softmax Transformers. That is, the attention weights are more concentrated (even sparsely) on some key features/tokens after training, which is observed in several existing works [Li et al., 2023a, Li et al., 2023b, Oymak et al., 2023] for softmax Transformers. Can the authors provide a comparison empirically on this? Also, I think it is better to cover such a discussion in the revision.

Oymak et al., 2023, "On the Role of Attention in Prompt-tuning. "
Li et al., 2023a, "A Theoretical Understanding of Shallow Vision Transformers: Learning, Generalization, and Sample Complexity."
Li et al., 2023b, "How do transformers learn topic structure: Towards a mechanistic understanding."

---

> ### Author Response · Authors · 2023-11-17
> **Response**
>
> Thank you for appreciating the values of our work and your constructive comments!
>
>
> Admittedly, we do not provide theoretical understanding of the phenomenon, but what we claim as our main contribution is the discovery of a simple abstract setting where the key characteristics of Transformer optimization persists, and we believe that our findings could serve as a useful testbed for the future theoretical investigations.
>
> In order to address your concern regarding the limitation, we conducted an additional set of experiments for a nonlinear regression. In particular, we consider the case where the covariates $x^{(i)}$ are distorted by a ReLU network $MLP$, i.e., the responses are generated as $y^{(i)} = \langle w_\star , MLP(x^{(i)}) \rangle$. In order to cope with the nonlinearity we also added a MLP in our Transformer architecture.
> As you can see from **Appendix B**, the results for the nonlinear regression are largely similar to the linear regression. We hope that our additional set of experiments addresses some of your concern.
>
> Moreover, thank you for point us to the additional relevant works regarding the softmax Transformers. Although it's not directly related to the main message of our paper, we answer your question regarding the attention scores as below. As a summary, for the linear Transformer for linear regression, we need to combine appropriate covariates $x^{(i)}$, $i=1,2,...,n$ to figure out the missing prediction $y^{(n+1)}$ for $x^{(n+1)}$. So, the sparsity of the attention weights will depend on instances. For instance, when the covariates $x^{(i)}$ are orthogonal, we empirically observe that the attention scores are sparse both for linear and softmax Transformers, which is consistent with the results from the previous papers. As per your suggestion, we added a discussion regarding the softmax Transformer on **Page 6**.

---

### Meta-Review · Area_Chair_XVVS · 2023-12-05

**Metareview:**

This paper proposes a simple toy model of Transformer optimization (via shallow linear Transformers), which they show reproduces many relevant features of larger Transformers. This is an important contribution, since simpler toy models are more scientifically tractable to study. Linear models have been very successful as theoretical toy models for standard DNN learning, so it is encouraging to see them potentially relevant to Transformers as well.

I recommend the paper for acceptance. All reviewers are in support, and note that this is an insightful contribution to the community.
In the camera-ready, beyond addressing the remaining reviewer concerns, I expect the authors to add a much more thorough discussion of Limitations (including those brought up in the rebuttal phase). This will better situate this paper’s contributions.

**Justification For Why Not Higher Score:**

As reviewers noted, the experiments are small-scale and somewhat limited. Moreover, the paper is missing a clear discussion of the limitations of the model.

**Justification For Why Not Lower Score:**

It is important to develop tractable abstractions of real-world models. All reviewers are in support.

---

### Decision · Program_Chairs · 2024-01-16

Accept (poster)